# Expectation maximization based framework for joint localization and parameter estimation in single particle tracking from segmented images

Ye Lin[1], Sean B. Andersson[1,2]*

**1** Division of Systems Engineering, Boston University, Boston, MA, United States of America, **2** Department of Mechanical Engineering, Boston University, Boston, MA, United States of America

* sanderss@bu.edu

**Data Availability Statement:** All simulation data are available from the Dryad database (DOI: 10.5061/dryad.9w0vt4bf5).

## Abstract

Single Particle Tracking (SPT) is a well known class of tools for studying the dynamics of biological macromolecules moving inside living cells. In this paper, we focus on the problem of localization and parameter estimation given a sequence of segmented images. In the standard paradigm, the location of the emitter inside each frame of a sequence of camera images is estimated using, for example, Gaussian fitting (GF), and these locations are linked to provide an estimate of the trajectory. Trajectories are then analyzed by using Mean Square Displacement (MSD) or Maximum Likelihood Estimation (MLE) techniques to determine motion parameters such as diffusion coefficients. However, the problems of localization and parameter estimation are clearly coupled. Motivated by this, we have created an Expectation Maximization (EM) based framework for simultaneous localization and parameter estimation. We demonstrate this framework through two representative methods, namely, Sequential Monte Carlo combined with Expectation Maximization (SMC-EM) and Unscented Kalman Filter combined with Expectation Maximization (U-EM). Using diffusion in two-dimensions as a prototypical example, we conduct quantitative investigations on localization and parameter estimation performance across a wide range of signal to background ratios and diffusion coefficients and compare our methods to the standard techniques based on GF-MSD/MLE. To demonstrate the flexibility of the EM based framework, we do comparisons using two different camera models, an ideal camera with Poisson distributed shot noise but no readout noise, and a camera with both shot noise and the pixel-dependent readout noise that is common to scientific complementary metal-oxide semiconductor (sCMOS) camera. Our results indicate our EM based methods outperform the standard techniques, especially at low signal levels. While U-EM and SMC-EM have similar accuracy, U-EM is significantly more computationally efficient, though the use of the Unscented Kalman Filter limits U-EM to lower diffusion rates.

**Funding:** SBA 1R01GM117039-01A1 National Institutes of Health. The funders had no role in study design, data collection and analysis, decision to publish, or preparation of the manuscript.

**Competing interests:** The authors have declared that no competing interests exist.

## Introduction

Single particle tracking (SPT) is an important class of techniques for studying the motion of single biological macromolecules. With the ability to localize particles with an accuracy far below the diffraction limit of light and to track particles across time, SPT continues to be an invaluable tool in understanding biology at the nanometer-scale by revealing details about particle dynamics and their local environment such as diffusion rates, confinement length, and other parameters [1]. SPT has been applied to a wide variety of molecules, including proteins [2, 3], mRNA molecules [4], DNA [5], viruses [6, 7], growth factor receptor [8], Janus colloids [9], and more [10].

Typically, SPT analysis of a sequence of images begins with an image segmentation step where the raw images are post-processed to extract image sequences that each contain information about a single particle. These sequences are then further processed to determine particle trajectories and motion model parameters. Under the standard paradigm, a two-step process is applied. In the first step, the location of the particle in each segmented image frame is determined and linked across frames to form a trajectory [11] (we refer to this as "localization refinement" since the initial segmentation is a coarse localization step). In the second step, trajectories are analyzed to extract information about the dynamic process, such as the value of the diffusion coefficient or other motion parameters. Localization refinement is often done using Gaussian Fitting (GF) [12, 13] while model parameters are extracted from the trajectories using the Mean Square Displacement (MSD) [14, 15] or Maximum Likelihood Estimation (MLE) [16, 17]. Regardless of the algorithms used, this two-step paradigm separates trajectory estimation from model parameter identification despite the fact that these two problems are coupled.

One of the assumptions of the standard approach is that the localized positions represent a simple linear observation of the true particle position corrupted by additive white Gaussian noise. The actual data, however, are usually the segmented camera images. The photon detection process in each pixel during imaging can be well modeled as a Poisson-distributed random variable with a rate that depends on the true location of the particle as well as on experimental realities, including background intensity noise and the details of the optics used in the instrument. This already nonlinear model becomes even more complicated at the low signal intensities that are often found in SPT data where noise models specific to the type of camera being used become relevant, whether it be a Charge-Coupled Device (CCD), Electron Multiplying CCD (EMCCD), Complementary Metal-Oxide Semiconductor (CMOS), or scientific CMOS (sCMOS) device [18, 19].

To handle nonlinear measurement models and to simultaneously estimate localization and parameter estimation, one of the authors introduced an approach based on nonlinear system identification [20]. This general approach, known as *Sequential Monte Carlo-Expectation Maximization* (SMC-EM), can handle nearly arbitrary nonlinearities in both the motion and observation models and has been shown to work as well as current state-of-the-art methods in the simple settings of 2-D diffusion and to work under more complicated motion and observation scenarios including estimating 3-D motion from wide field images. However, one drawback of this approach is its computational complexity due to the use of a particle filter and a particle smoother to handle those nonlinearities. Recently, we addressed this issue by replacing the particle-based methods with an Unscented Kalman filter (UKF) and an Unscented Rauch-Tung-Striebel smoother (URTSS) [21], a scheme we refer to as *Unscented-Expectation Maximization* (U-EM). Compared to the SMC-EM approach, U-EM significantly decreases the computational load, allowing the method to be applied to larger data sets and to more complicated models. This reduction in complexity comes, however, at the cost of generality in the

posterior distribution describing the position in the particle at each time point since the UKF-URTSS approximates this distribution as a Gaussian while the SMC-EM can represent essentially arbitrary distributions [22].

Because SPT experiments are often photon-impoverised and subject to significant background, it is important to consider the impact of signal and noise levels when comparing different analysis algorithms. For example, [23] investigated the performance of an experimental method in error estimation techniques across a variety of signal and noise values, the comparison work in [11] included the signal level as a core factor in their simulations, [24] generated simulated videos at various levels of signal to noise ratios to validate the use of convolutional neural networks on SPT data, and [25] applied deep learning to analyze particle trajectories based on simulated data over a large range of signal to noise ratios. Though the standard SPT methods perform well at high signal levels, many begin to fail as the signal level decreases or noise level increases. This motivates us to compare our EM based methods (e.g., SMC-EM and U-EM) to the standard methods (e.g., GF-MSD and GF-MLE) across a wide variety of Signal-to-Background (SBR) levels. The noise sources considered in this work consist of three parts: the shot noise inherent to any photon detection process, the background noise arising from out-of-focus fluorescence or auto-fluorescence, and the readout noise of the camera sensor. The readout noise depends on the type of camera being used, e.g., the readout noise from sCMOS/CMOS cameras is pixel-dependent, while the architecture of CCD/EMCCD sensors allows every single pixel be treated in the same way. Here we consider two different camera models, an ideal camera model with shot and background noise only, and an sCMOS camera model with shot, background, and pixel-dependent readout noise.

In order to validate these algorithms against a known ground-truth, we carried out quantitative comparisons of our EM based methods and the standard SPT methods through extensive simulations under the assumption that segmentation has already been performed. For concreteness, we focus on a fairly simple setting where GF-MSD and GF-MLE are known to work well, namely that of 2-D diffusion. Our algorithms can be readily extended to 3-D tracking, including those based on different imaging modalities such as confocal schemes [26], as well as to more complicated motion models (such as Ornstein-Uhlenbeck motion, directed motion, and other Markovian models), and a variant has recently been developed for analyzing SPT data with time-varying parameters [27].

There are two primary contribution of this work. The first is the extension of our existing algorithms to data captured using an sCMOS camera. Due to their relatively low cost, high speed, and performance, sCMOS cameras are becoming popular tools for SPT data acquisition and including them in our EM-based approach extends the impact our algorithms can have. The second is the detailed, quantitative comparison of our EM based methods to a standard in the field, namely GS-MSD, and to an existing alternative that is also based on optimal estimation theory and which has previously been shown to outperform the standard approach in the analysis of diffusion, namely GF-MLE. This comparison is done across a wide range of SBRs and across a wide range of diffusion coefficients, validating the performance of our methods and guiding users in algorithm selection based on their particular experimental conditions.

## Methods

In this section, we provide a brief introduction to the techniques considered in this work.

### Localization then parameter estimation

In the standard two-step approach, illustrated in Fig 1, the raw images are first segmented and then these segmented images are processed to yield localizations of the particle. Next, the

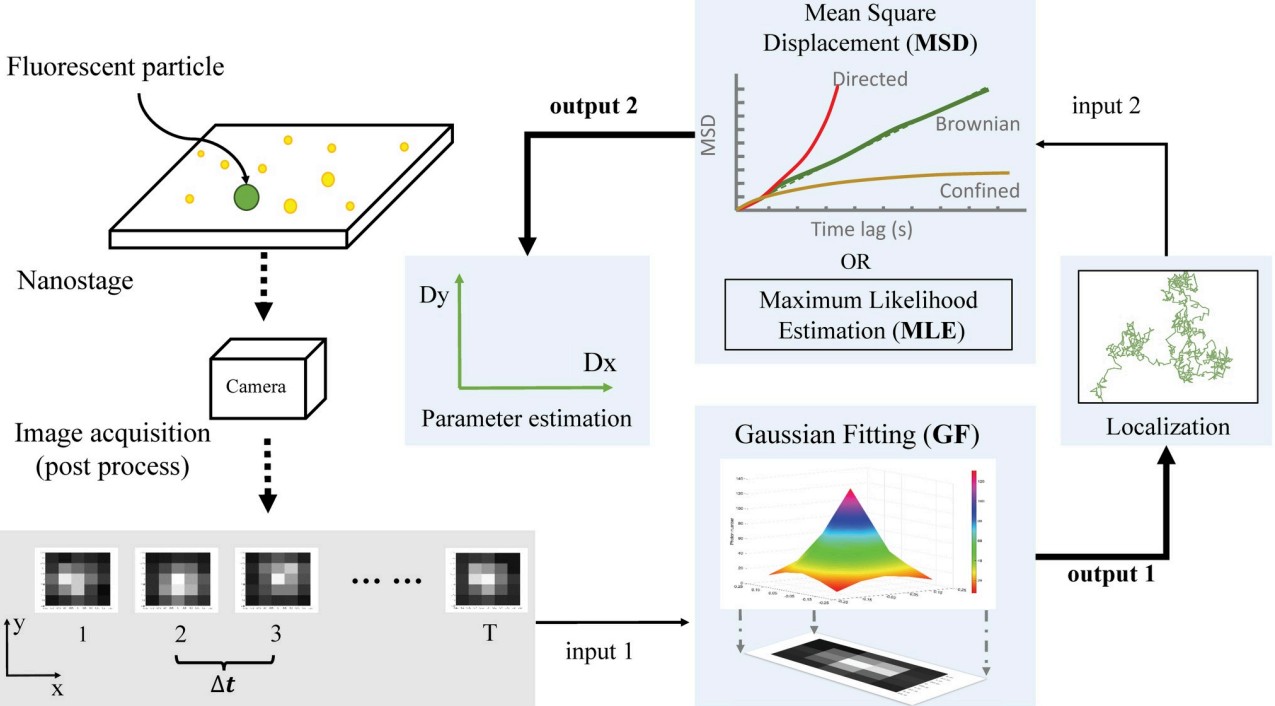

**Fig 1. Illustration of the standard approach.** Segmented image data is first passed through a localization step where an algorithm such as GF determines the position of the particle in each frame. The resulting trajectory is then analyzed using, e.g., the MSD or MLE to determine model parameters.

resulting trajectory is analyzed to estimate motion model parameters. While there are a variety of localization algorithms, we focus here on Gaussian fitting as it remains a popular approach due to its simplicity and accuracy, particularly in the 2-D setting.

**Gaussian Fitting (GF).** In the 2-D setting, the Point Spread Function (PSF) of the instrument is well approximated by a Gaussian. As a result the measured intensity, $I_{xy}$ can be described as

$$I_{xy} = G \exp\left(-\frac{(x-x_o)^2}{2\sigma_x{}^2} - \frac{(y-y_o)^2}{2\sigma_y{}^2}\right) + N_{bgd}, \qquad (1)$$

where $G$ is the peak amplitude of the intensity, $(x, y)$ are the lateral coordinates in the image frame, $(x_o, y_o)$ are the position of the particle, $(\sigma_x, \sigma_y)$ are physical parameters describing the width of the PSF, and $N_{bgd}$ is the background intensity. Fitting the measured data to this model allows one to estimate the particle position as well as the other model parameters in Eq (1).

**Mean Square Displacement (MSD).** MSD is one of the most frequently used methods for estimating diffusion coefficients from trajectory data. Following [14], the single-axis MSD is given by

$$\text{MSD}(n) = \frac{1}{N-n} \sum_{i=1}^{N-n} (r_{i+n} - r_i)^2, \quad n = 1, \ldots, N-1, \qquad (2)$$

where $N$ is the data length and $r_i$ is the position of the fluorescent particle in either $x$ or $y$ in frame $i$. For a particle moving with a diffusion coefficient of $D$, the expectation of the MSD is

given by

$$\mathbb{E}[\text{MSD}(n)] = 2Dn\Delta t, \tag{3}$$

where $\Delta t$ is the time interval between frames of the image sequence. In this work we fit calculated MSD curves to the model in Eq (3) using the nonlinear least-squares curve fitting solver *lsqnonlin* in MATLAB (MathWorks, Natick, MA).

**Maximum Likelihood Estimation (MLE).**   While the MSD remains popular and is simple to use, it relies on several user choices and is known to lack robustness with respect to measurement noise. An alternative is to use optimal estimation theory. In particular, the MLE has good statistical properties as it is both efficient and consistent, achieving the Cramer-Rao lower bound (so long as sufficient data is available) [28]. Consider the problem of identifying an unknown parameter $\theta \in \mathbb{R}^{n_\theta}$ for an arbitrary state space model

$$X_{t+1} = f_t(X_t, w_t, \theta), \tag{4a}$$

$$Y_t = h_t(X_t, v_t, \theta), \tag{4b}$$

where $X_t$ is the (vector) state of the system at time $t$, $Y_t$ is the (vector) observation at time $t$, $w_t$ and $v_t$ are independent white noise processes, and $\theta_t$ is the unknown (vector) parameter to be estimated. The MLE determines an estimate of this parameter by maximizing the log likelihood of the observed data $Y_{1:N} \triangleq \{Y_1, \ldots, Y_N\}$,

$$\hat{\theta} = \arg\max_\theta \log p_\theta(Y_{1:N}), \tag{5}$$

where $p_\theta(Y_{1:N})$ is the joint probability density of the observations $Y_{1:N}$ defined by the model in Eq (4). We use a computationally efficient version of the ML estimator developed in [16] for estimating the diffusion coefficient and the variance of observation noise $v_t$ under the assumption of a simple diffusion motion model and a linear observation of the position corrupted by zero-mean Gaussian noise. The reader is referred to [16, 28] for details.

## Simultaneous localization and parameter estimation

The basic tool behind our approach of simultaneous localization and parameter estimation is the Expectation Maximization (EM) algorithm [22], an iterative approach for finding an ML estimate. Based on EM, we created a generic framework for SPT analysis shown in Fig 2. In what follows, we briefly describe our approach and the two flavors of it used in this work. Note that the EM approach does not produce a point estimate for the particle location in each frame but rather an estimate of the smoothed probability distribution of its location and thus provides more information than the GF. For the purposes of this work, we obtain the particle location by taking the mean of this distribution in each frame; however, other estimators could be used.

**Expectation Maximization (EM).**   Consider once again the state space model in Eq (4). In general, the log-likelihood of the observations, $\log p_\theta(Y_{1:N})$, is intractable or cannot be written analytically. As a result, Eq (5) cannot be solved directly. The EM algorithm handles this through an iterative approach, forming an approximation to the likelihood function at the $e^{th}$ step, named $\mathcal{Q}(\theta, \theta^{(e)})$, based on a current estimate of the parameter $\theta^{(e)}$, and then optimizing this to find the next estimate $\theta^{(e+1)}$, stepping towards the MLE [29]. The approximation is given by the conditional expectation of the joint log likelihood function,

$$\mathcal{Q}(\theta, \ \theta^{(e)}) = \mathbb{E}_{\theta^{(e)}}[L_\theta(X_{0:N}, Y_{1:N})|Y_{1:N}], \tag{6}$$

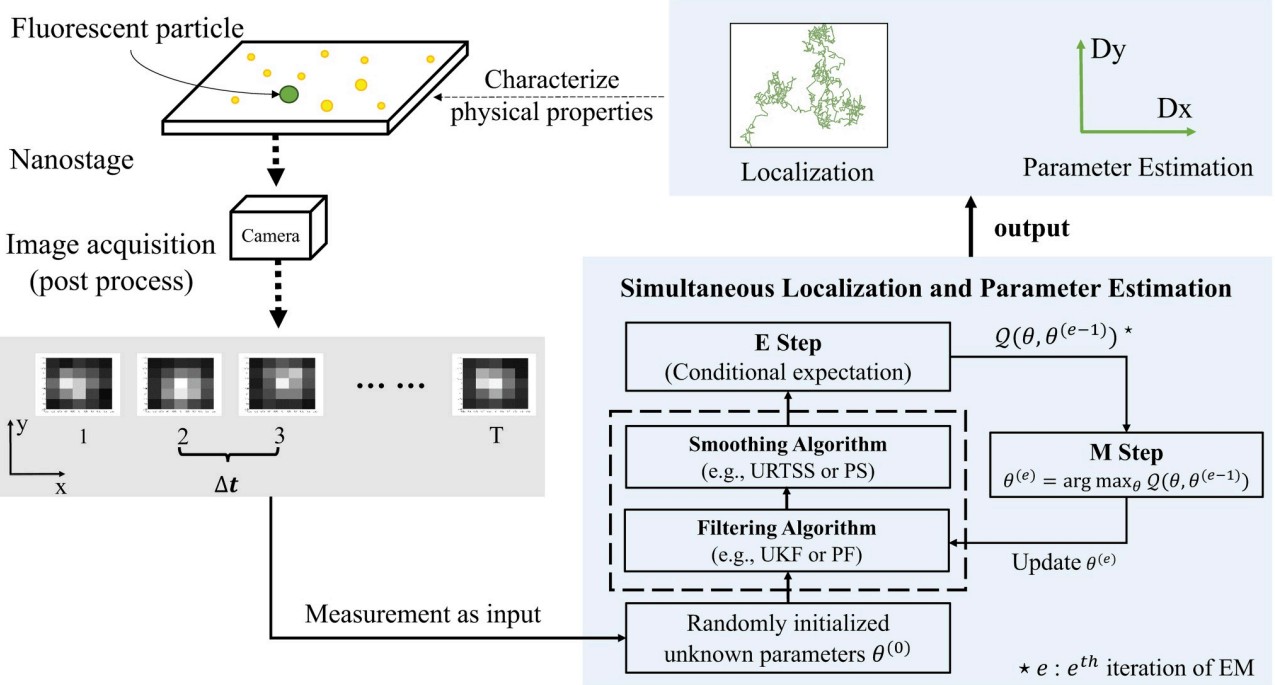

**Fig 2. Illustration of the EM-based framework for simultaneous localization and parameter estimation.** Segmented image data is passed directly to the estimation routine where EM alternates between filtering/smoothing to find the distribution of the particle trajectory and estimation of the parameter based on that distribution.

where $\theta$ is the unknown parameter, $X_{0:N} = \{X_0, X_1, \cdots, X_N\}$ is known as a *hidden state* that, in the context of SPT, is given by the unknown particle locations, and $L_\theta(X_{0:N}, Y_{1:N})$ is the joint log likelihood function of the trajectory and observations. This function is given by

$$L_\theta(X_{0:N}, Y_{1:N}) = \log p_\theta(X_0) + \sum_{t=1}^{N} \log p_\theta(X_t|X_{t-1}) + \sum_{t=1}^{N} \log p_\theta(Y_t|X_t). \tag{7}$$

Using Eq (7) in Eq (6) yields

$$\mathcal{Q}(\theta, \ \theta^{(e)}) = I_1(\theta, \theta^{(e)}) + I_2(\theta, \theta^{(e)}) + I_3(\theta, \theta^{(e)}), \tag{8}$$

where

$$I_1(\theta, \theta^{(e)}) = \mathbb{E}[\log p(X_0|\theta)|Y_{1:N}, \theta^{(e)}], \tag{9a}$$

$$I_2(\theta, \theta^{(e)}) = \sum_{t=1}^{N} \mathbb{E}[\log p(X_t|X_{t-1})|Y_{1:N}, \theta^{(e)}], \tag{9b}$$

$$I_3(\theta, \theta^{(e)}) = \sum_{t=1}^{N} \mathbb{E}[\log p(Y_t|X_t)|Y_{1:N}, \theta^{(e)}]. \tag{9c}$$

The calculation of $\mathcal{Q}(\theta, \theta^{(e)})$ is called the *Expectation (E) step* at the $e^{th}$ iteration. It has been shown that any choice of $\theta^{(e+1)}$ such that $\mathcal{Q}(\theta^{(e+1)}, \theta^{(e)}) \geq \mathcal{Q}(\theta^{(e)}, \theta^{(e)})$ ensures the EM algorithm converges to a local maximum of the likelihood function. Thus, the expectation step is

followed by a *Maximization (M) step* to produce the next estimate of the parameter,

$$\theta^{(e+1)} = \arg\max_{\theta} \mathcal{Q}(\theta, \theta^{(e)}) \tag{10}$$

Despite the fact that convergence is only guaranteed to a local optimum, EM has been shown to work well in practice [29, 30]. To implement the *E* step (that is, to calculate $\mathcal{Q}$) by carrying out the expectations in Eq (9), it is necessary to know the posterior densities $p(X_t|Y_{1:N})$ and $p(X_t, X_{t-1}|Y_{1:N})$. If the underlying model in Eq (4) is linear with Gaussian noise, then these distributions are easily obtained using a Kalman filter and a Kalman smoother [31]. For nonlinear observations, however, these distributions must often be approximated in some way. Here, we apply two approaches as described below.

**Unscented—EM (U-EM).**   U-EM approximates the posterior densities in Eq (9) as Gaussians using an Unscented Kalman Filter (UKF) and an Unscented Rauch-Tung-Striebel Smoother (URTSS). The UKF was developed in [32] as an alternative to the Extented Kalman Filter, capturing (an approximation to) the mean and covariance of a nonlinear stochastic process without relying on linearization or a Jacobian computation. U-EM starts with the UKF to get the estimated state and covariance, and then uses the URTSS to return the posterior probability densities required for the EM algorithm. U-EM is significantly more computationally efficient than the Monte Carlo scheme SMC-EM described below. However, it relies on a few hand-tuned parameters and, as will be seen in Case 3 of the simulation studies below, its performance suffers at large diffusion coefficients. There are variants that may offer superior performance (see, e.g. [33]) but with some additional complexity. The details of U-EM are presented in the S1 Text.

**Sequential Monte Carlo—EM (SMC-EM).**   For the SMC-EM algorithm, the posterior densities in Eq (9) are calculated using a Particle Filter (PF) and Particle Smoother (PS), allowing for arbitrary distributions to be estimated. Note that the term "particles" in SMC refers to the random samples used to represent a distribution rather than the fluorescently labeled objects being tracked. In the remainder of the paper, the meaning of the word "particles" should be clear from context. The details of SMC-EM can be found in the S2 Text.

## Simulation and results

In this section, we describe the motion and observation models in the scenario where a subdiffraction-sized particle is imaged with a widefield fluorescence microscope. We then describe simulation studies of the algorithms comparing their performance against each other and against GF-MSD and GF-MLE under two different camera models. We begin by considering an ideal camera, modeling the Poisson distributed shot noise common to all photon detection processes. While we include background noise and a basic model of a pixelated image, we ignore other camera-specific issues such as readout noise. Under this scenario we consider three cases. In Case 1, we evaluate the performance of the algorithms in an experimental setting with a relatively large signal and a low background. Case 2 then studies algorithm performance across a range of signal intensity and background levels. In Case 3, we explore the effect of the value of the diffusion coefficient on algorithm performance, considering both an idealized setting with no motion blur and a more realistic setting where motion blur is presented. In the second scenario, we include pixel-dependent readout noise to capture the behavior of sCMOS camera sensors. Under this scenario we consider two different SBRs when comparing algorithm performance.

## Ground truth simulation

**Motion model.**   We assume the fluorescent particle moves according to a simple Brownian diffusion. The state dynamics $f_t(\cdot)$ in Eq (4a) are then given by

$$X_{t+1} = X_t + W_t \quad W_t \sim \mathcal{N}(0, Q),$$ (11)

where $X_t$ is a column vector representing the location of the fluorescent particle in the lateral plane at time $t$, and $Q$ is a covariance matrix given by

$$Q = \begin{bmatrix} 2D_x \Delta t & 0 \\ 0 & 2D_y \Delta t \end{bmatrix},$$ (12)

where $D_x$ and $D_y$ are independent diffusion coefficients in each of the coordinate axes and $\Delta t$ is the time interval between frames of the image sequence. Note that in general, the anisotropy of the diffusion is not necessarily aligned to the coordinate axes and the cross-correlation term in the symmetric matrix $Q$ should also be estimated. Including this additional term in the EM-based estimation methods is straightforward; the MSD and MLE methods, however, assume either isotropic diffusion or independent axes.

**Observation model.**   Because the single particle is smaller than the diffraction limit of light, the image on the camera is described by the PSF of the instrument. In 2-D (and in the focal plane of the objective lens), the PSF is well approximated by

$$PSF(x, y; x_o, y_o) = G \cdot \exp\left(-\frac{(x - x_o)^2}{2\sigma_x^2} - \frac{(y - y_o)^2}{2\sigma_y^2}\right), \quad \sigma_x = \sigma_y = \frac{\sqrt{2}\lambda}{2\pi \text{NA}},$$ (13)

where $(x_o, y_o)$ is the location of the particle, $G$ is the peak intensity of the fluorescence, $\lambda$ is the wavelength of the emitted light and NA is the numerical aperture of the objective lens being used [34].

Assuming segmentation has already been done, the image acquired by the camera is composed of $P^2$ pixels arranged into a $P \times P$ square array. The pixel size is $\Delta x$ by $\Delta y$ with the actual dimensions determined both by the physical size of the camera elements on the camera and by the magnification of the optical system. At time step $t$, the expected photon intensity measured for the $p^{th}$ pixel, $\lambda_{p,t}$, is given by

$$\lambda_{p,t} = \int_{x_{p,t}^{min}}^{x_{p,t}^{max}} \int_{y_{p,t}^{min}}^{y_{p,t}^{max}} \frac{1}{\Delta x \Delta y} PSF(\xi, \xi'; x_t, y_t) \, d\xi d\xi',$$ (14)

where $(x_t, y_t)$ is the position of the particle, and the integration bounds $(x_{p,t}^{min}, x_{p,t}^{max})$ and $(y_{p,t}^{min}, y_{p,t}^{max})$ are over the boundaries of a given pixel.

In addition to the signal, there is always a background intensity rate arising from out-of-focus fluorescence and autofluorescence in the sample. This can be modeled locally as a uniform rate $N_{bgd}$ over the small $P \times P$ array of the segmented images [20]. Usually, the value of the backgroud noise is measured experimentally and for the rest of this paper, we assume it is known (though its value can be estimated using the EM algorithm). Due to the shot noise inherent to the photon generation process, the measured intensity in each pixel is given by a Poisson process [35] with the value in the $p^{th}$ pixel at time $t$ given by

$$I_{p,t} \sim \text{Poiss}(\lambda_{p,t} + N_{bgd}) + \epsilon_{p,t},$$ (15)

where $\text{Poiss}(\cdot)$ represents a Poisson distribution and $\epsilon_{p,t}$ denotes the readout noise. To model an ideal camera, we take $\epsilon_{p,t} = 0$, while for an sCMOS camera the readout noise depends on the

particular pixel. (Details on the specific pixel characteristics used in this work can be found in the S3 Text.) The final observation vector $Y_t$ in the model Eq (4b) is then the collection of all $P^2$ pixel values.

This model is used to generate all the data in the simulations as well as for the SMC-EM calculations. However, the UKF algorithm inside the U-EM scheme assumes measurements are corrupted by additive noise. It is thus necessary to transform the Poisson distributed model in Eq (15) into one with additive noise as a pre-calculation step before passing the measurements into the U-EM method. There are several ways to achieve this for a Poisson-distributed random varaible, including directly approximating it as a Gaussian random variable or by using a variance-stabilising transformation such as an Anscombe or Freeman Tukey transformation. In prior work we compared these methods and found that in general the Anscombe transformation produced the best results [21]. This method transforms a Poisson-distributed random variable into a Gaussian one with the same mean but unity variance [36, 37]. To achieve this, the measurement data $I_{p,t}$ is transformed to $\tilde{I}_{p,t}$ according to

$$\tilde{I}_{p,t} = 2\sqrt{I_{p,t} + \frac{3}{8} + \sigma_{p,t}^2}, \tag{16}$$

where $\sigma_{p,t}^2$ is the variance of the readout noise in pixel $p$ at time $t$. The measurement model in (15) for computations in EM based framework is then replaced by

$$I_{p,t} \simeq 2\sqrt{\lambda_{p,t} + \frac{3}{8} + \sigma_{p,t}^2} + v_k, \quad v_k \sim \mathcal{N}(0,1). \tag{17}$$

For the purposes of the MSD and basic ML estimation, the observation model is a simple linear observation with Gaussian noise [15, 16, 28].

**Simulation setup.**   Simulations were made of a particle diffusing in 2-D, imaged for $N = 100$ frames at an imaging period of $\Delta t = 100$ ms (i.e, a frame rate of 10 frames/s) for a total of 10 s. To generate each sequence of images, independent trajectories of length $N \times N_{sub}$ were generated from the 2-D diffusion model Eq (11) where $N_{sub}$ represents a sub-sampling factor. In practice, cameras accumulate photons over an integration period, and the motion of the particle during the exposure period may affect the estimation accuracy. To replicate this motion blur effect, we assumed the camera accumulated photons continuously during the first $\delta t = 10$ ms of each imaging period $\Delta t$ and produced each final frame by averaging the first 10 consecutive images in the period and ignoring the rest. A typical image at a low signal level (here, $N_{bgd} = 1$, $G = 10$) is shown in the left-side image of Fig 3, while an image at a higher signal level ($N_{bgd} = 10$, $G = 100$) is shown in the right-side image. To generate statistics on algorithm performance, $K = 100$ image sequences were generated for every parameter setting. The fixed parameters used in the simulations are given in Table 1. All simulations and calculations were carried out using MATLAB.

## Ideal camera model

**Case 1: Performance at high signal and low background levels.**   For this first case, the peak signal level was set to $G = 100$ and the background noise to $N_{bgd} = 10$. (Note that these are the rates in each image after accumulating over the shutter period.) Other imaging parameters were set as described in Table 1. The diffusion coefficients were fixed to $D_x = 0.005 \ \mu m^2/s$ and $D_y = 0.01 \ \mu m^2/s$. The data were analyzed using GF-MSD, GF-MLE, U-EM, and three versions of SMC-EM: SMC$^{100}$, SMC$^{500}$, and SMC$^{1000}$ where the superscript denotes the number of Monte Carlo samples used in the PF and PS algorithms. A typical trajectory, together with the

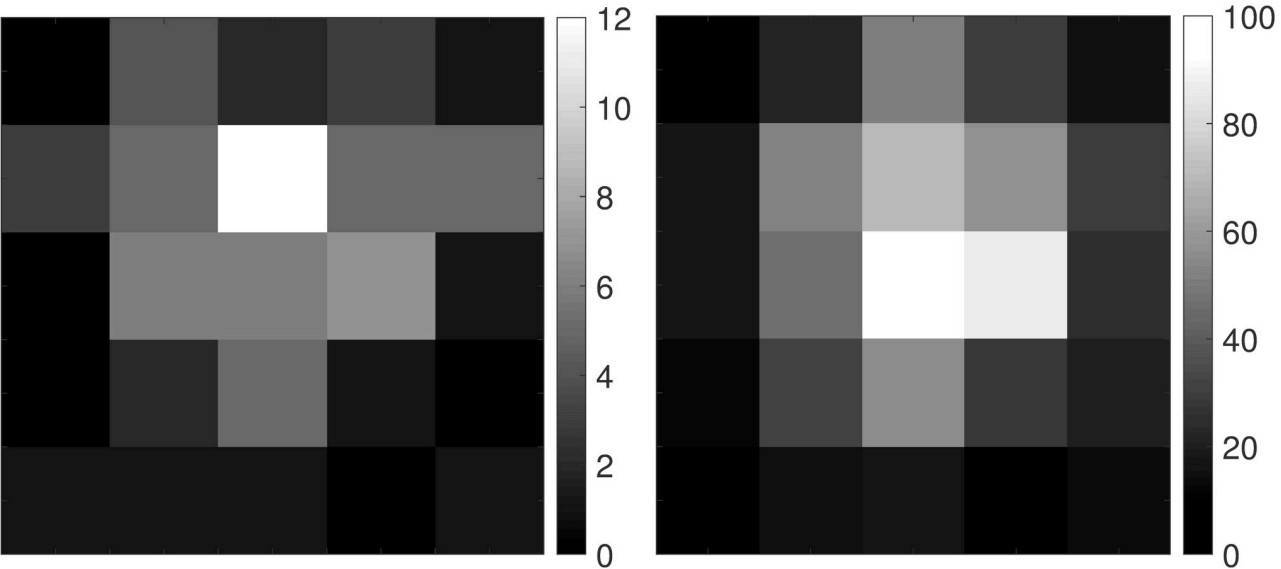

**Fig 3. Typical images at low and high signal levels.** (left) $N_{bgd} = 1$ and $G = 10$, (right) $N_{bgd} = 10$ and $G = 100$. Notice the different scaling in the two images.

position estimates produced by the U-EM algorithm as a typical estimation result, is shown in Fig 4.

As described in the methods section, the EM algorithm at the heart of the simultaneous approach is an iterative scheme, improving the estimate at each iteration as it moves towards a local optimal of the log likelihood function. The resulting evolution of the diffusion coefficient parameter estimates over the 100 different trajectories for the three versions of SMC-EM and for U-EM are shown in the box plots in Fig 5. These plots show that the EM algorithm generally converges in a small number of steps and that, as expected, the performance of SMC-EM improves as the number of Monte Carlo samples used in the PF and the PS increases.

The comparison between the final results across the 100 trajectories for all the algorithms are shown in the box plots in Fig 6 and recapitulated in Table 2. Results in $y$ are similar and are omitted for space reasons. Note that there is a clear bias in the diffusion coefficient estimation in the GF-MLE and in our EM based methods. This is likely driven by a variety of factors, including the length of the data set (since MLE methods are only guaranteed to be consistent, meaning that they converge to the true value as the amount of data becomes large) and nonlinearities in the models. A close examination of the SMC-EM results shows that bias reduces as we go from SMC-EM$^{100}$ to SMC-EM$^{500}$ and then to SMC-EM$^{1000}$. The SMC techniques more faithfully represent the nonlinear nature of the system as the number of MC samples. By contrast, the UKF at the heart of U-EM is accurate only to second-order. This supports the

**Table 1. Fixed parameters used in the simulations.**

| Symbol | Parameter | Value | Symbol | Parameter | Value |
|---|---|---|---|---|---|
| NA | Numerical aperture | 1.2 | $\lambda$ | Emission wavelength | 540 nm |
| $\Delta t$ | Imaging period | 100 ms | $\delta t$ | Shutter period | 10 ms |
| $P$ | Number of pixels | 25 | $\Delta x, \Delta y$ | Effective pixel length | 100 nm |
| $N$ | Image sequence length | 100 | $N_{sub}$ | Sub-sampling factor | 100 |
| $E$ | Number of SMC-EM iterations | 10 | $K$ | Number of sequences | 100 |

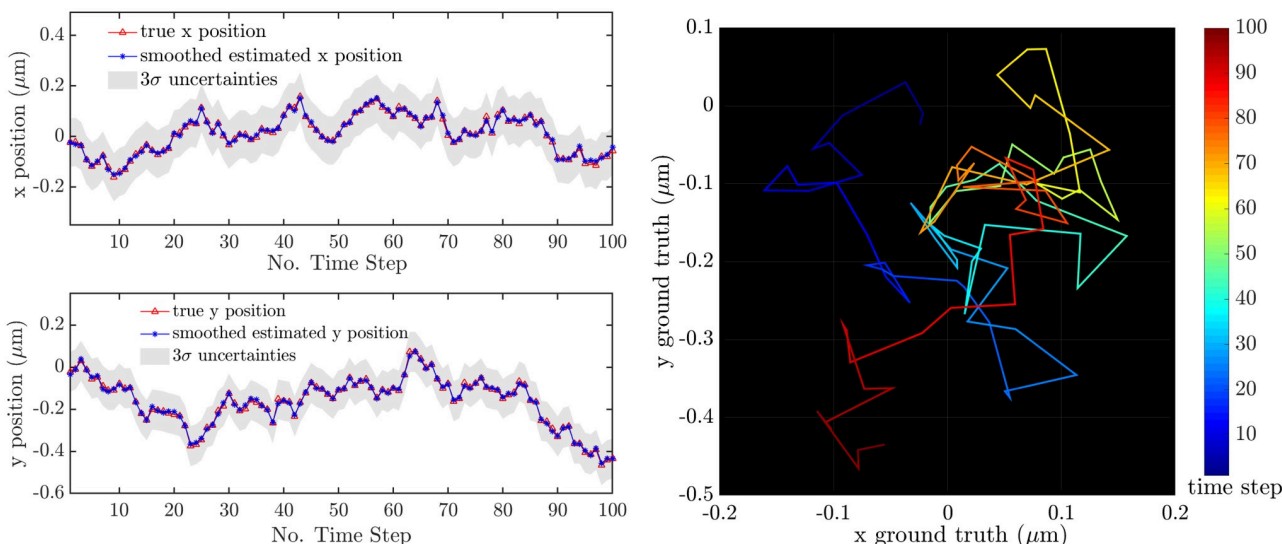

**Fig 4. A typical trajectory with $D_x$ = 0.005 $\mu$m²/s and $D_y$ = 0.01 $\mu$m²/s.** (left) $x$ and $y$ trajectories together with the position estimates from U-EM and the $3\sigma$ error bounds. (right) The ground truth trajectory in the plane with color indicating time.

argument that the nonlinearities in the observation models are at least partially responsible for driving the bias.

These results show that at these high signal levels, GF-MLE, SMC-EM, and U-EM all perform similarly in terms of diffusion coefficient estimation,. GF-MSD, however, while having a similar mean, has a much larger variance and many more outliers than the others. Localization performance is evaluated in terms of the Root Mean Squared Error (RMSE) over an entire trajectory. Both GF and the EM-based schemes yield accurate localization with mean errors of

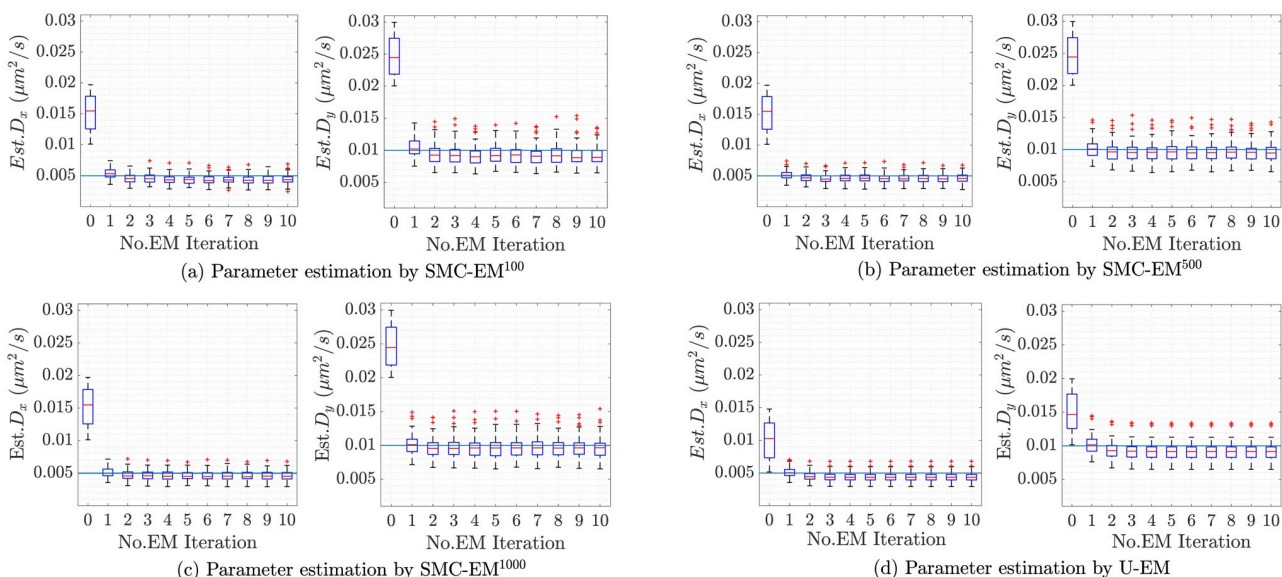

**Fig 5. Box plots of estimated $D_x$ and $D_y$ by SMC-EM and U-EM.** The true values of the diffusion coefficients are shown as solid horizontal lines in each plot. Note that the red line inside the box is the median, the edges of the box represent the first and third quartiles, the vertical dashed line indicates the bounds for data within 1.5 times the interquartile range, and the red + symbols are data points outside this range.

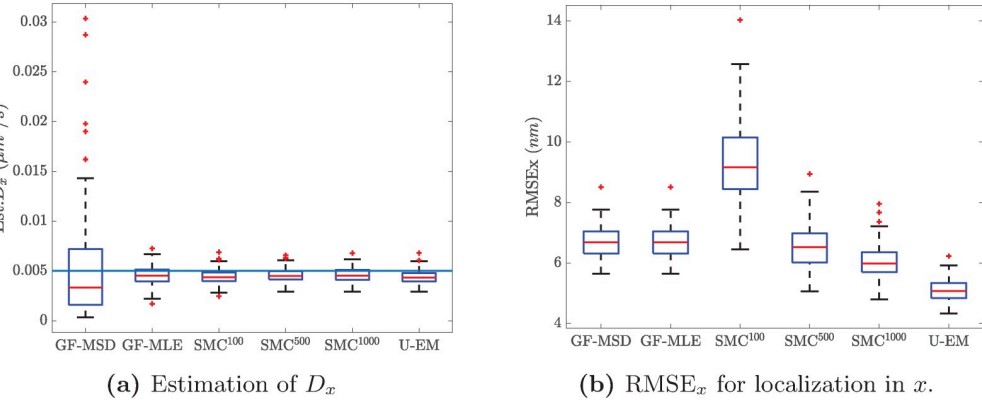

**Fig 6. Performance comparison among the different analysis methods.** With $G = 100$, $N_{bgd} = 10$, $D_x = 0.005$ $\mu m^2/s$, and $D_y = 0.01$ $\mu m^2/s$. (a) Box plot results for the estimate of $D_x$. (b) RMSE for $x$–localization.

below 7 nm for all but SMC$^{100}$ where the small number of Monte Carlo samples used to represent the location distribution leads to both a larger error and a larger variance relative to the other schemes. Table 2 also shows that the performance improvement is minimal when the number of Monte Carlo samples increases from 500 to 1000. In the remainder of this work, then, we use 500 particles in SMC-EM.

**Case 2: Performance at different signal and background noise levels.** In this second case, the diffusion coefficients were again fixed at $D_x = 0.005$ $\mu m^2/s$, $D_y = 0.01$ $\mu m^2/s$ and the imaging parameters set as in Table 1. The peak intensity, $G$, was varied across two decades, from $1 - 100$, and the background noise, $N_{bgd}$, was varied from 1 to 15. As before, 100 datasets of 100 images each were simulated at each pair of $\{G, N_{bgd}\}$ and the performance of the four algorithms, GF-MSD, GF-MLE, SMC$^{500}$, and U-EM compared.

*Parameter estimation performance.* To evaluate the parameter estimation performance among the different approaches, we followed the approach set out in [28] and defined a successful estimate as one which was within 25% of its true value. The success maps for each of the algorithms are shown in Fig 7. In these plots, color corresponds to the percentage of runs where successful estimation was achieved. Results for $D_y$ were similar and are omitted for space reasons. These results show that GF-MSD has the worst performance of all four algorithms across all settings of intensity and background noise level with very low rates of success even at the highest SBR and signal levels considered. At the absolute lowest signal levels, GF-MLE shows the highest success rate (though that rate is still very low). The two EM-based

**Table 2. Algorithm performance at $G = 100$, $N_{bgd} = 10$, $D_x = 0.005$ $\mu m^2/s$, $D_y = 0.01$ $\mu m^2/s$.**

| Approach | Est.$D_x$ ($\mu m^2/s$) | Est.$D_y$ ($\mu m^2/s$) | RMSE$_x$ (nm) | RMSE$_y$ (nm) |
|---|---|---|---|---|
| GF-MSD | $0.0055 \pm 0.0059$ | $0.0102 \pm 0.0096$ | $6.7 \pm 0.524$ | $6.7 \pm 0.506$ |
| GF-MLE | $0.0046 \pm 9.72e\text{-}4$ | $0.0092 \pm 0.0019$ | $6.7 \pm 0.524$ | $6.7 \pm 0.506$ |
| SMC-EM$^{100}$ | $0.0044 \pm 7.76e\text{-}4$ | $0.0092 \pm 0.0015$ | $9.2 \pm 1.100$ | $9.8 \pm 1.300$ |
| SMC-EM$^{500}$ | $0.0046 \pm 7.23e\text{-}4$ | $0.0097 \pm 0.0015$ | $6.6 \pm 0.757$ | $6.5 \pm 0.629$ |
| SMC-EM$^{1000}$ | $0.0046 \pm 7.15e\text{-}4$ | $0.0097 \pm 0.0015$ | $6.0 \pm 0.580$ | $5.9 \pm 0.488$ |
| U-EM | $0.0044 \pm 7.0013e\text{-}4$ | $0.0091 \pm 0.0013$ | $6.3 \pm 0.475$ | $7.7 \pm 1.200$ |

Note that the estimates in table are in the form of *mean ± Std.* while the boxplots in Fig 6 indicate the median.

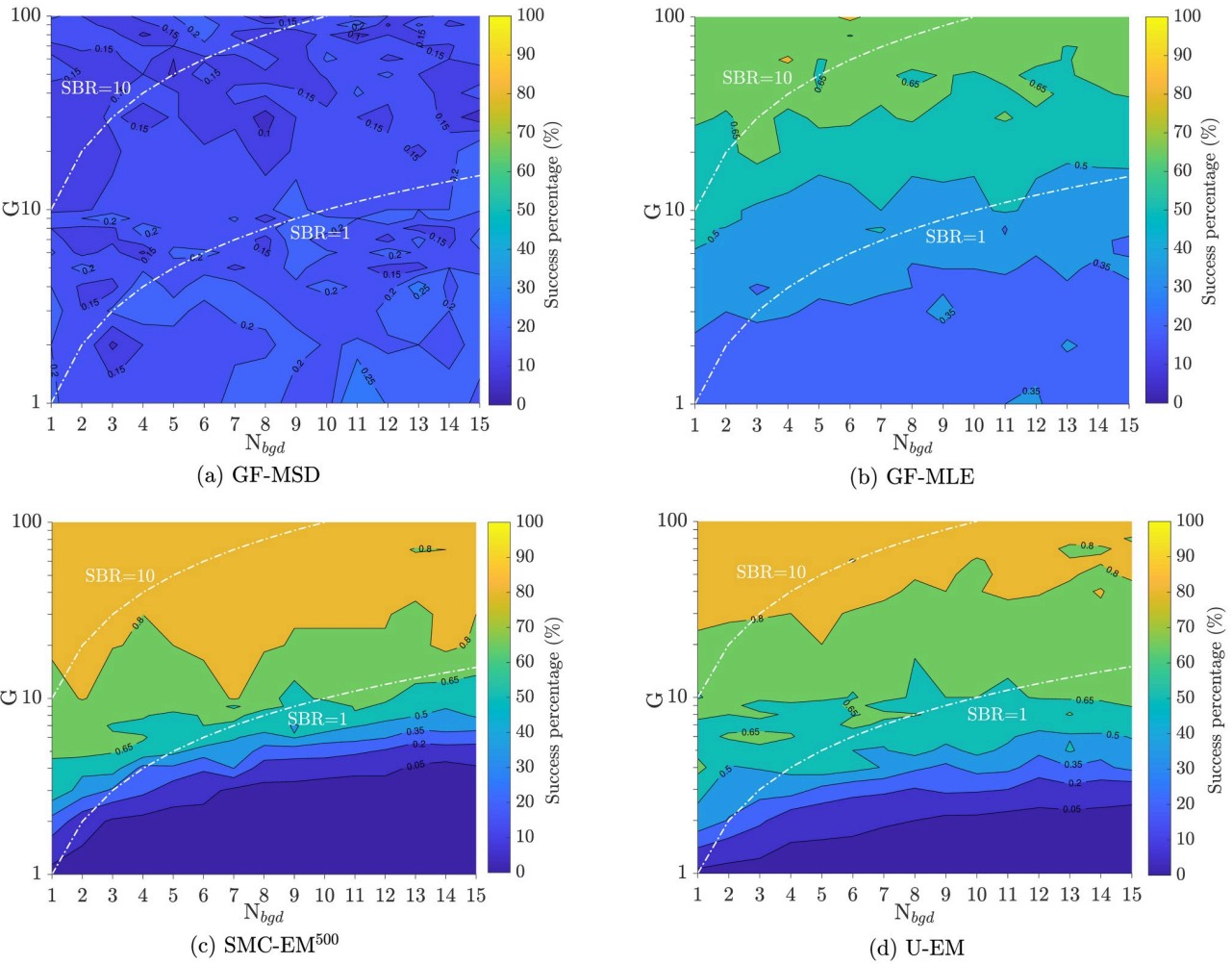

**Fig 7. Success maps of the four algorithms.** The percentage of trajectories resulting in an estimated $D_x$ within 25% of the true value as a function of peak intensity $G$ and background level $N_{bgd}$ is shown. Yellow indicates 100% success while blue represents 0%. Results along the two white curves are shown in Fig 8.

methods, however, show the highest level performance when the entire range of SBRs is considered.

To dive more deeply into these results, we compared the performance in the accuracy of diffusion coefficient estimation at SBRs of $G/N_{bgd} = 10$ and $G/N_{bgd} = 1$ along the two white curves on the success maps in Fig 7. The parameter estimation results are shown in Fig 8 for both SBR = 1 (representative case at low signal intensities) and SBR = 10 (representative case at high signal intensities). The figures show the mean and median for all algorithms as well as the middle two quantile (50%) range. Note that at SBR = 1, the GF-MSD approach essentially fails while GF-MLE needs an intensity of $G = 10$ before its estimates are reasonable. By contrast, our EM based methods return reasonable results beginning at an intensity of $G = 3$. With an SBR of 10, all four algorithms yield reasonable results, though the GF-MSD algorithm has the worst performance with a median value that significantly under-reports relative to the true value and with quantiles that are much larger than those of the other schemes. The other three algorithms all have similar performance, though the EM-based methods do yield tighter quantiles.

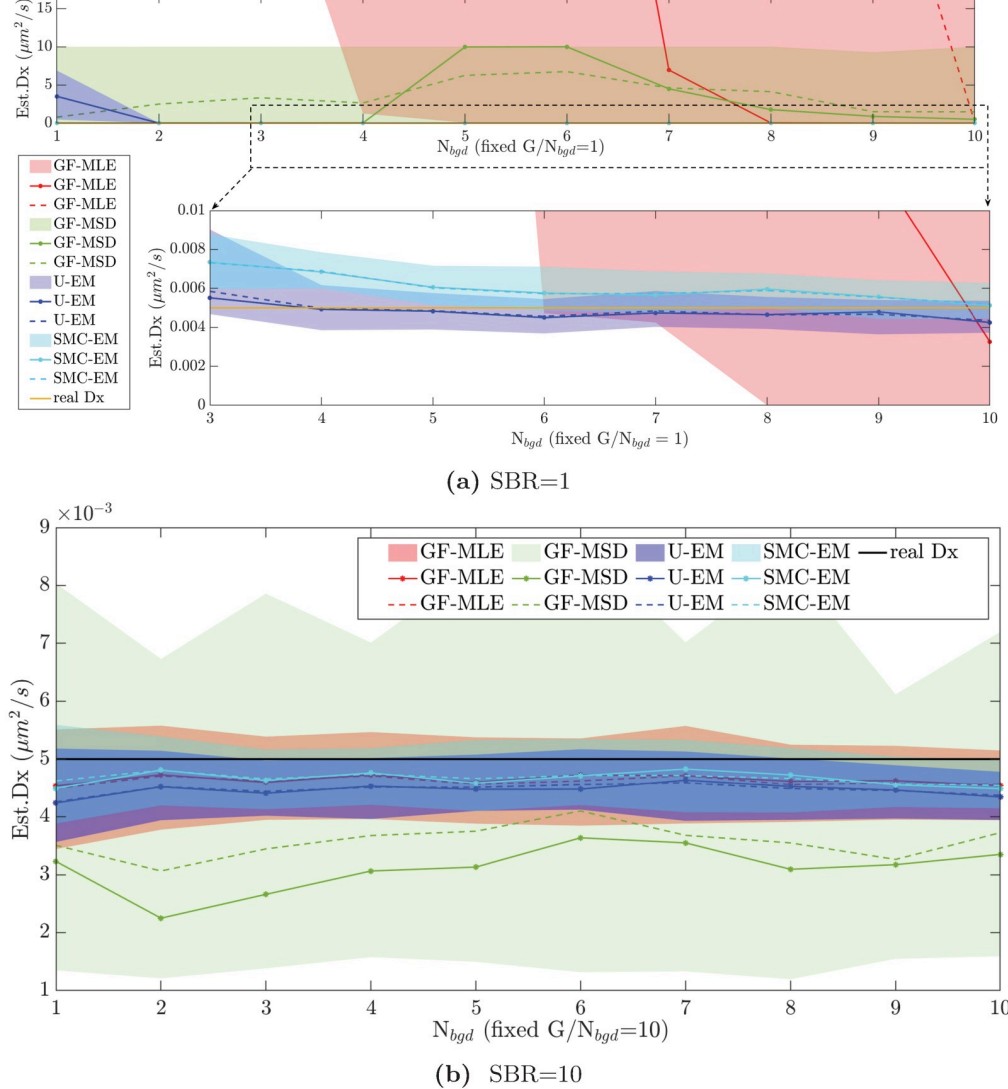

**Fig 8. Diffusion coefficient estimation performance over 100 simulation runs at different signal and background levels.** With fixed (a) SBR = 1 and (b) SBR = 10. Shown are the middle two quantiles (colored, shaded area), median (solid line), and mean (dashed line) using GF-MLE (red), GF-MSD(green), U-EM (purple), and SMC-EM$^{500}$ (cyan). The true value was $D_x = 0.005 \, \mu\text{m}^2/\text{s}$.

*Localization performance*. We also compared the localization accuracy of the different algorithms for the same data. The results for both SBR = 1 and SBR = 10 are shown in Fig 9. As before, we show the center two quantiles, mean, and median of the estimates over the 100 trials at each value of $G$ and $N_{bgd}$. (Note that since both the GF-MSD and GF-MLE algorithms use GF for localization, their results are combined as they are equivalent.) SMC-EM$^{500}$ outperforms the other algorithms at all signal levels. Except at the very lowest signal level, U-EM outperforms GF. As the signal level increases, GF eventually catches up to match the results of SMC-EM and U-EM.

**Case 3: Performance as a function of diffusion coefficient.** In the presence of motion blur, the performance of both localization and parameter estimation will depend on the diffusion coefficient. In addition, because the EM-based schemes jointly estimate the trajectory and

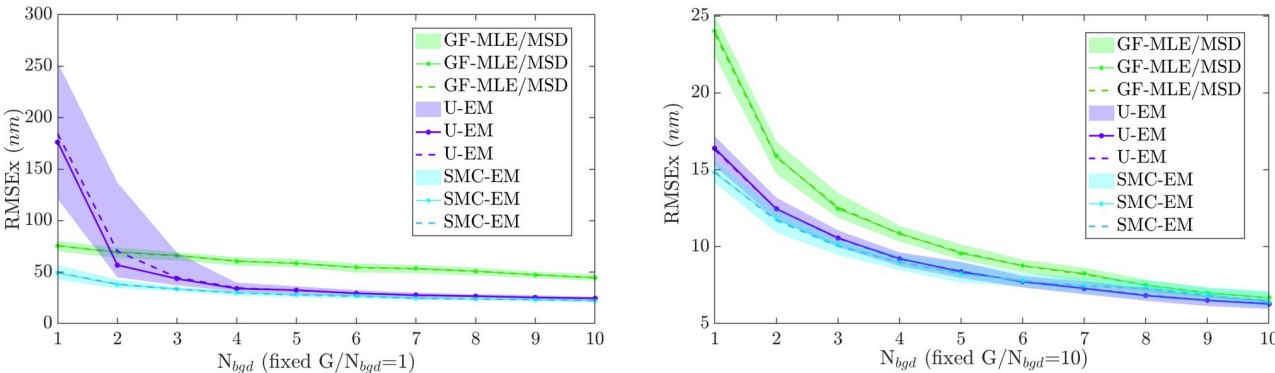

**Fig 9. Localization performance (RMSE) over 100 simulation runs at different signal and background levels.** With fixed (left) SBR = 1 and (right) SBR = 10. Shown are the middle two quantiles (colored, shaded area), median (solid line), and mean (dashed line). GF (green), U-EM (purple), and SMC-EM$^{500}$ (cyan).

the model parameters, it is reasonable to expect that performance will depend on motion model parameters even in the absence of motion blur (representing the limit of instantaneous image acquisition). To study this, we fixed the signal levels at $G = 100$, $N_{bgd} = 10$ (where all algorithms perform similarly) and ran simulations with $D_x = D_y$ over the range of 0.001 $\mu m^2$/s to 10 $\mu m^2$/s, considering both the case with motion blur (with $N_{sub} = 100$) and without (with $N_{sub} = 1$). We set a threshold for localization failure as the diffraction limited resolution given by the Rayleigh criterion. For the imaging parameters used here this leads to

$$\Delta L_{Rayleigh} = \frac{0.61\lambda}{NA} = 270 \, \text{nm}.$$

The results for localization in the absence of motion blur are shown in Fig 10a. As expected, if the measurements can be obtained instantaneously then the performance of the GF method is independent of the diffusion coefficient since in each frame the particle is motionless. The EM-based schemes, however, do show degraded performance as the diffusion coefficient increases with the resulting thresholds shown in Table 3. It is perhaps somewhat surprising that U-EM and SMC-EM have such drastically different thresholds given that they use the same observation model. However, the problem in U-EM arises primarily from the the breakdown of the unscented transform at large noise variances. Thus, while U-EM is much better than SMC-EM in terms of computational complexity, it is limited to small values of the ratio

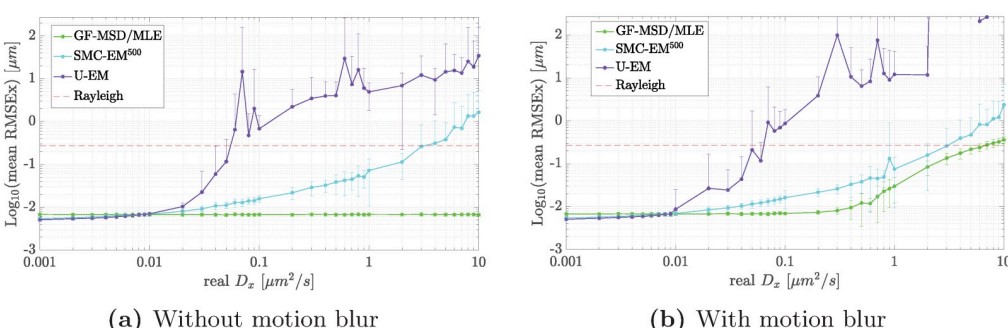

**Fig 10. Localization performance in terms of RMSE across varying diffusing speeds.** (a) without and (b) with motion blur using GF-MSD/MLE (green), SMC-EM$^{500}$ (cyan), and U-EM (purple). The failure threshold is defined as the Rayleigh resolution criterion (red, dashed).

**Table 3. Diffusion coefficient threshold before localization failure with $G = 100$, $N_{bgd} = 10$.**

| Condition | GF ($\mu$m²/s) | U-EM ($\mu$m²/s) | SMC-EM$^{500}$ ($\mu$m²/s) |
|---|---|---|---|
| No motion blur | $\infty$ | 0.05 | 3.0 |
| With motion blur | 6.0 | 0.05 | 3.0 |

of diffusion coefficient to sample rate due to the inability of the UKF to handle large variances in the noise inputs.

The case with motion blur is shown in Fig 10b. U-EM and SMC-EM perform similarly to the setting without motion blur. Now, however, estimation based on GF also shows a limit on the diffusion coefficient beyond which localization fails, likely driven by the fact that motion blur causes the PSF to diverge from a simple Gaussian shape. The resulting limits are shown in Table 3. It is important to note that when using GF, we take advantage of the prior information available in the segmentation and limit the estimate to be within the segmented image. To give the EM scheme maximum flexibility, we do not do this for SMC-EM or U-EM.

The values of the thresholds for the diffusion coefficient depend, of course, on the specific imaging parameters. In general, in the absence of motion blur, increasing the peak intensity $G$ or the shutter time $\delta t$ will increase the SBR and thus improve localization performance and one would expect the SMC-EM methods to work at higher diffusion coefficient values. Since U-EM depends on the unscented transform, increasing the imaging rate (that is, decreasing $\Delta t$) will reduce the process noise and thus increase the diffusion coefficient threshold. In the presence of motion blur, decreasing the shutter time will mitigate its effects but at the cost of reducing the number of acquired photons and thus the SBR. This can be compensated for somewhat by increasing the intensity parameter $G$ by increasing the power of the excitation, though the ability to do so is limited by phototoxicity issues.

To better understand the degradation in localization as the diffusion coefficient increases, we show in Fig 11 typical runs both with and without motion blur. These results show that at larger diffusion coefficients, the U-EM scheme simply fails while the others degrade more smoothly, particularly in the absence of motion blur. For SMC-EM, we show results using different numbers of sampled particles in the PF methods. With a small number of samples (e.g., SMC-EM$^{100}$) and at large $D$, the SMC-EM tends to track the particle well in most frames but occasionally to lose that track. Because segmentation ensures that the data is never too far from ground truth, the algorithm is often able to pick up the location again. Increasing the number of sampling particles in the SMC-EM mitigates this effect at any given value of $D$ and thus the threshold on the diffusion coefficient increases with increasing number of particles in the filter. To further demonstrate this, we also show typical results when using 1500 sampling particles; SMC-EM is then able to track the particle even at 10 $\mu$m²/s.

The results for the estimation of $D_x$ as a function of the diffusion coefficient are shown in Fig 12, both with and without motion blur. As noted before, the UKF element of the U-EM algorithm fails as the covariance of the process noise, defined by the value of the diffusion coefficient, gets large. As seen in the localization performance results in Fig 11, this leads to complete loss of tracking which in turn leads to failed diffusion coefficient estimation. By contrast, SMC-EM, GF-MSD, and GF-MLE continue to produce good estimates throughout the entire considered range (though, of course, GF-MSD has much larger variance than the other approaches. It is perhaps surprising that SMC-EM produces good diffusion coefficients at large $D$ even though, as seen in Fig 10, localization performance degrades significantly. The likely reason is that, as illustrated in Fig 11, for a large part of any given trajectory, tracking is

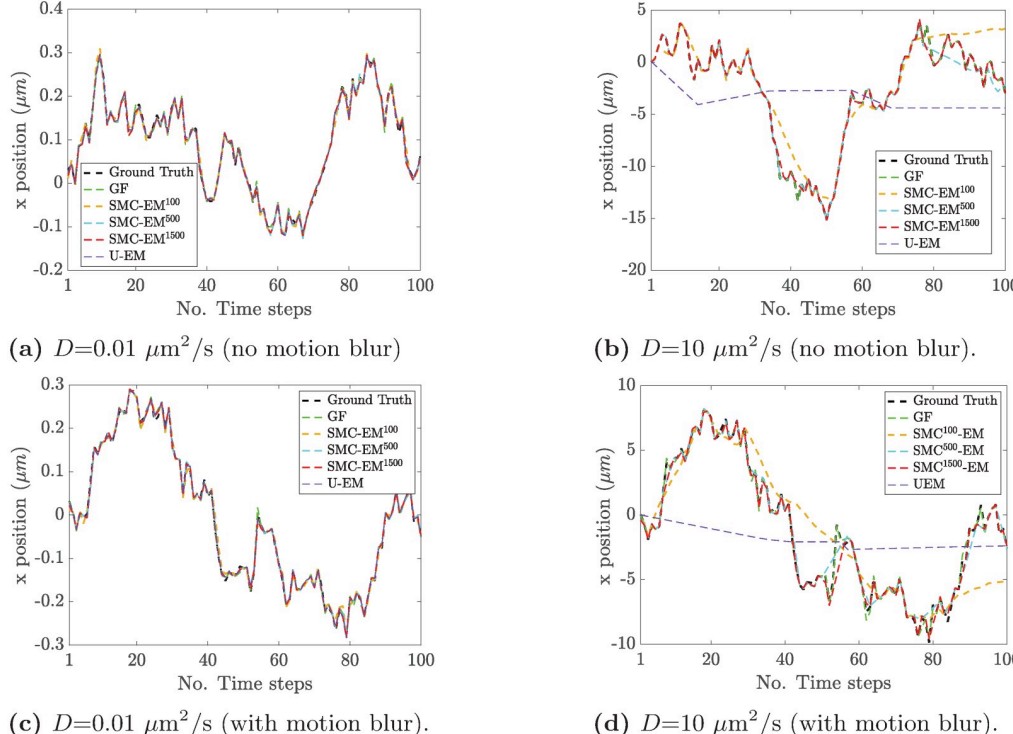

**Fig 11. Typical localization performance of GF, SMC-EM, and U-EM.** (green) GF, (orange) SMC-EM with 100 Monte Carlo samples, (cyan) 500 Monte Carlo samples, (red) 1500 Monte Carlo samples, and (purple) U-EM at a diffusion coefficient of (a,c) 0.01 $\mu$m$^2$/s and (b,d) 10 $\mu$m$^2$/s, both (a,b) without and (c,d) with motion blur.

good with only a few large outliers. These outliers have a serious impact on the RMSE but a smaller effect on the diffusion coefficient.

## sCMOS camera model

To simulate an sCMOS camera, we include pixel-dependent readout noise in the measurement model through the choice of distributions for $\epsilon_{p,t}$ in Eq (15). We base our measurement model on a Hamamatsu ORCA Flash 4.0 camera described in [38]; details can be found in the S3 Text. The corresponding probability density function (PDF) of the measured photon counts in

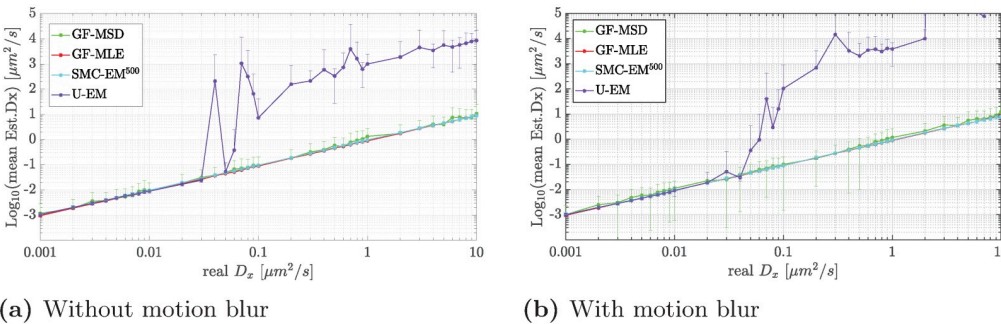

**Fig 12. Mean estimates of $D_x$ by GF-MSD, GF-MLE, SMC-EM, and U-EM as a function of the true diffusion coefficient.** (a) without and (b) with motion blur.

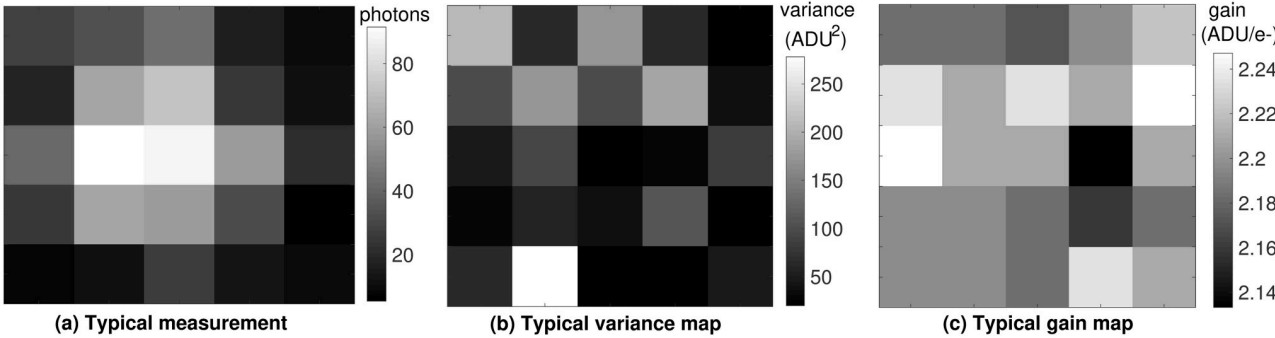

**Fig 13. Typical frames related to sCMOS camera model.** (a) Observation with $N_{bgd} = 10$, $G = 100$. (b) Variance and (c) gain maps of the pixels in the frame shown in (a).

pixel $p$ at time $t$ is given by

$$P(I_{p,t}) = \sum_{q=0}^{\infty} \frac{1}{q!} \exp\left[-(\lambda_{p,t} + N_{bgd})\right](\lambda_{p,t} + N_{bgd})^q \frac{1}{\sqrt{2\pi\sigma_{p,t}^2}} \exp\left[-\frac{(I_{p,t} - q)^2}{2\sigma_{p,t}^2}\right]. \tag{18}$$

Derivation of this distribution can be found in the S4 Text.

Simulations were performed using the settings in Table 1 and at two different signal levels, one low ($G = 10$) and one high ($G = 100$). A typical image frame at $G = 100$ and $N_{bgd} = 10$, together with the pixel-by-pixel variance and gain maps for this frame, is shown in Fig 13. A video of a typical image sequence can be found in the S1 Video. The work in [38] showed that GF can yield poor results on sCMOS data and, motivated by this, developed a localization algorithm specific to the sCMOS model using ML estimation. In the remainder of this work, then, we use that approach to localize the particle in each frame. For easy reference, details of this algorithm can be found in the S5 Text. A comparison of the computation time for all the algorithms can also be found in the S1 Fig. We combine those localization results with the MLE approach to parameter estimation from [16] and refer to this combined algorithm as MLE_sCMOS+.

The comparison between the final results across all 100 simulation runs for all the algorithms at the high signal level are shown in the box plots in Fig 14 and recapitulated in Table 4. These results indicate that when the signal level is high, all methods perform well in parameter estimation and localization, though when using only 100 particles in SMC-EM the RMSE is higher than with the other methods. In addition, SMC-EM$^{500}$ and SMC-EM$^{1000}$ show fewer outliers than the other methods.

The comparison between the final results across all 100 simulation runs for all the algorithms for the low signal level are shown in the box plots in Fig 15 and recapitulated in Table 5. The EM-based schemes show significant improvement over MLE_sCMOS in this setting in terms of both reduced variance in the parameter estimates and smaller RMSE in localization.

To further highlight the localization performance differences between the algorithms, in Fig 16 we show the results from a typical run. While all methods track the true trajectory, MLE_sCMOS produces more outliers while the EM-based methods stay closer to the trajectory throughout the run.

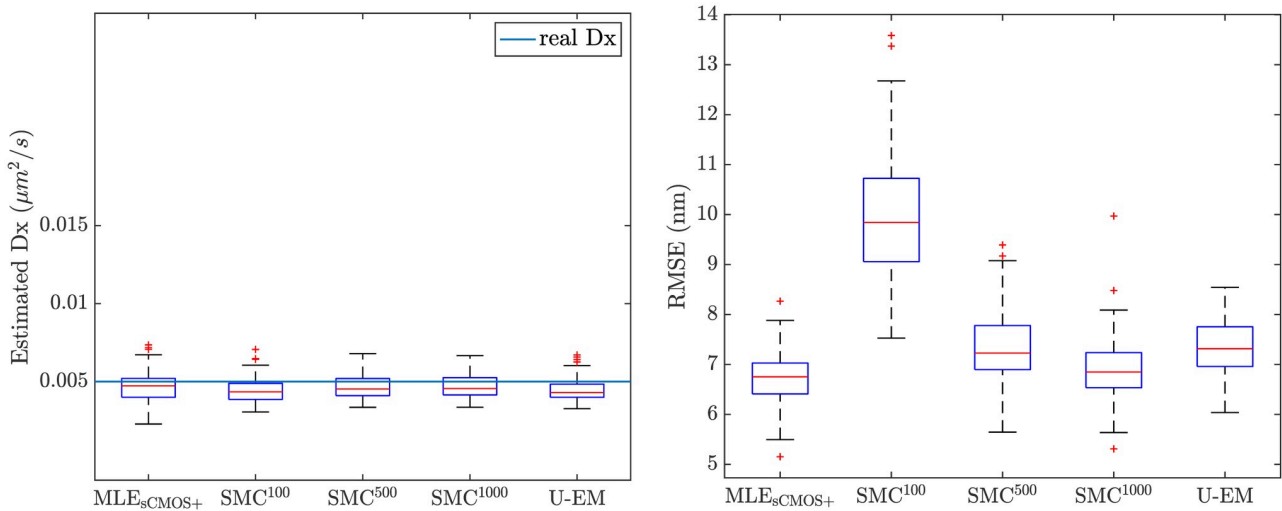

**Fig 14. Estimation performance of different SPT methods on sCMOS camera model at $G = 100$ and $N_{bgd} = 10$.**

**Table 4. Algorithm performance at $G = 100$, $N_{bgd} = 10$, $D_x = 0.005$ $\mu$m$^2$/s, $D_y = 0.01$ $\mu$m$^2$/s with pixel-dependent readout noise.**

| Approach | Est.$D_x$ ($\mu$m$^2$/s) | Est.$D_y$ ($\mu$m$^2$/s) | RMSE$_x$ (nm) | RMSE$_y$ (nm) |
|---|---|---|---|---|
| MLE$_{sCMOS+}$ | 0.00472 ± 0.00105 | 0.00922 ± 0.00199 | 6.72 ± 0.513 | 6.82 ± 0.514 |
| SMC-EM$^{100}$ | 0.00450 ± 0.00084 | 0.00947 ± 0.00142 | 9.97 ± 1.338 | 11.19 ± 1.857 |
| SMC-EM$^{500}$ | 0.00470 ± 0.00079 | 0.00964 ± 0.00144 | 7.35 ± 0.711 | 7.65 ± 0.756 |
| SMC-EM$^{1000}$ | 0.00473 ± 0.00078 | 0.00968 ± 0.00145 | 6.90 ± 0.620 | 7.04 ± 0.560 |
| U-EM | 0.00448 ± 0.00074 | 0.00894 ± 0.00135 | 7.36 ± 0.547 | 8.99 ± 1.22 |

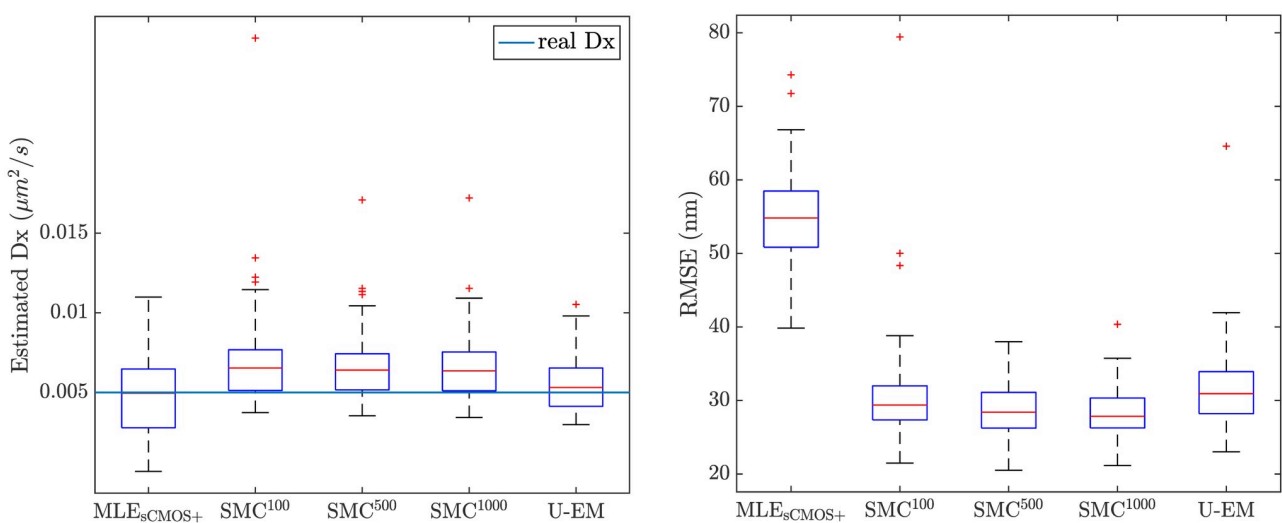

**Fig 15. Estimation performance of different SPT methods on sCMOS camera model at $G = 10$, $N_{bgd} = 10$.**

**Table 5. Algorithm performance at $G = 10$, $N_{bgd} = 10$, $D_x = 0.005$ $\mu m^2/s$, $D_y = 0.01$ $\mu m^2/s$ with pixel-dependent readout noise.**

| Approach | Est.$D_x$ ($\mu m^2/s$) | Est.$D_y$ ($\mu m^2/s$) | RMSE$_x$(nm) | RMSE$_y$ (nm) |
|---|---|---|---|---|
| MLE$_{sCMOS+}$ | 0.00484 ± 0.00247 | 0.00999 ± 0.00391 | 54.59 ± 6.18 | 54.76 ± 6.45 |
| MLE$^*_{sCOMS+}$ | 0.00484 ± 0.00247 | 0.00999 ± 0.00391 | 54.22 ± 5.65 | 54.55 ± 6.12 |
| SMC-EM$^{100}$ | 0.00699 ± 0.00287 | 0.0114 ± 0.00312 | 30.77 ± 6.65 | 36.50 ± 9.65 |
| SMC-EM$^{100, *}$ | 0.00661 ± 0.00177 | 0.0112 ± 0.00286 | 29.89 ± 3.61 | 35.23 ± 4.30 |
| SMC-EM$^{500}$ | 0.00668 ± 0.00206 | 0.0110 ± 0.00311 | 28.71 ± 3.47 | 33.92 ± 4.49 |
| SMC-EM$^{500, *}$ | 0.00642 ± 0.00159 | 0.0109 ± 0.00294 | 28.71 ± 3.47 | 33.33 ± 3.31 |
| SMC-EM$^{1000}$ | 0.00662 ± 0.00207 | 0.0109 ± 0.00307 | 28.46 ± 3.46 | 33.44 ± 4.04 |
| SMC-EM$^{1000, *}$ | 0.00646 ± 0.00171 | 0.0108 ± 0.00291 | 28.34 ± 3.26 | 33.02 ± 3.26 |
| U-EM | 0.00556 ± 0.00165 | 0.00931 ± 0.00293 | 31.58 ± 5.26 | 44.00 ± 17.24 |
| U-EM$^*$ | 0.00551 ± 0.00158 | 0.00891 ± 0.00239 | 31.25 ± 4.08 | 38.56 ± 6.52 |

$^*$ excluding outliers.

## Discussion and conclusion

In this work we described two versions of our EM-based framework for simultaneous localization and parameter estimation from SPT data. We extended them to include an observation model describing sCMOS cameras and compared their performance in terms of localization and diffusion coefficient estimation to GF-MSD and GF-MLE for an ideal camera and to MLE$_{sCMOS}$ for an sCMOS camera. Our algorithms indicate that, at least in the two-dimensional setting considered, if there are enough photons and a good SBR, then GF-MLE (or MLE$_{sCMOS}$), SMC-EM, and U-EM perform similarly well and all outperform GF-MSD. Given the additional computational complexity of the EM-based methods over GF-MLE/MLE$_{sCMOS}$, it makes more sense to apply these more standard algorithms in this setting. At low signal levels, however, the EM-based methods outperform the others. The choice between the different

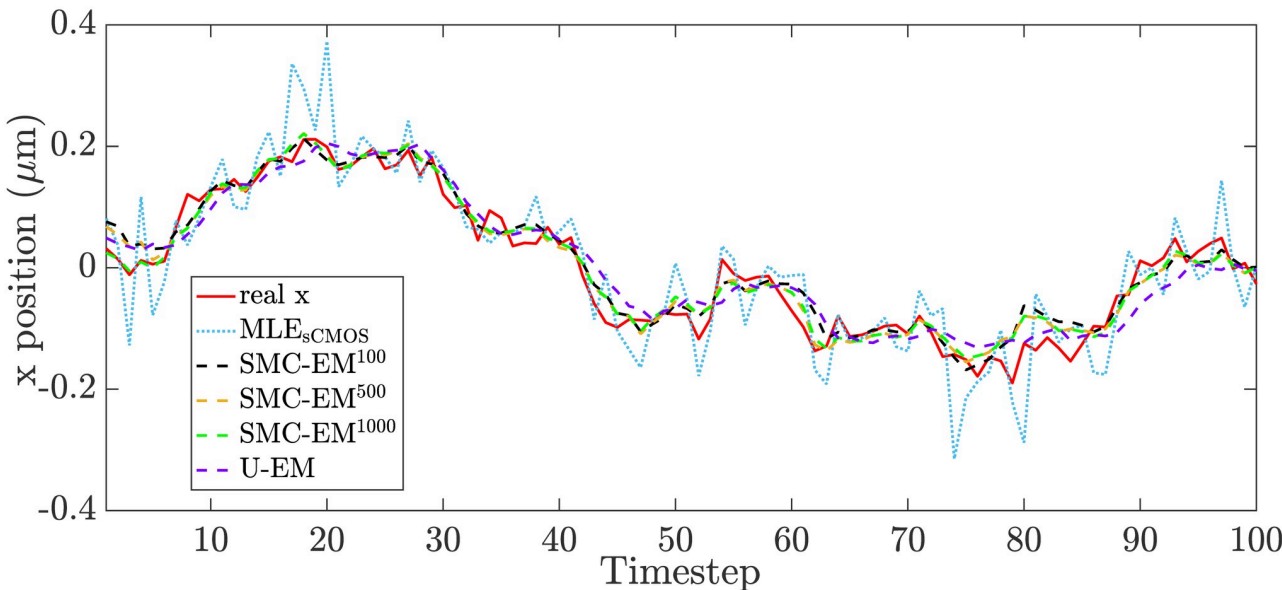

**Fig 16. Typical localization results at $G = N_{bgd} = 10$.**

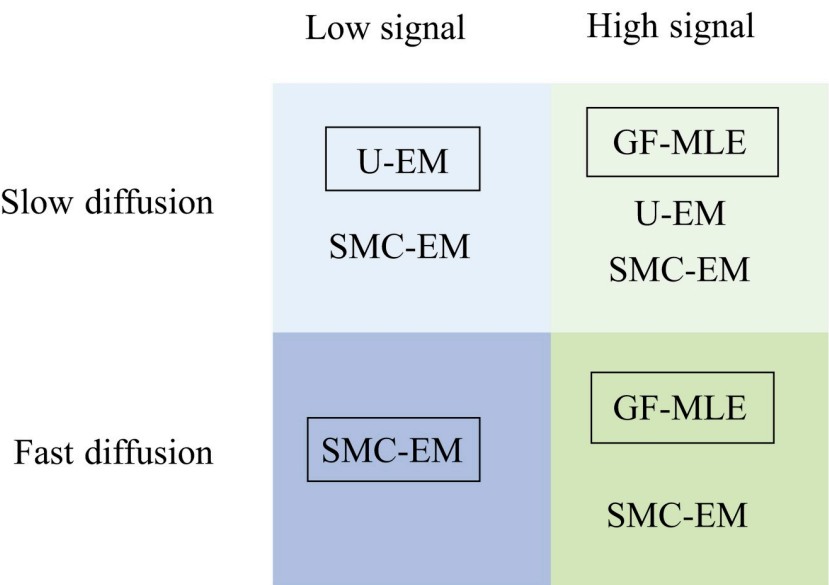

**Fig 17. Qualitative guidance for choice of SPT localization and parameter estimation algorithm.** Shown are the algorithms that produce similar results in each of the domain with the method. The boxed algorithm in each quadrant has the lowest computational load.

EM schemes is dictated in large part by the computation time but also by the ratio of the (expected) diffusion coefficient and the sampling rate. If this ratio is low, U-EM offers good results at significantly less computation time than the SMC-EM methods. These conclusions are summarized in Fig 17.

It is important to note that the EM-based methods are quite flexible and can be easily adapted to other measurement and motion models and that they return a full distribution for the position of the particle in each frame rather than a single point estimate. This additional information may be useful when asking, for example, the likelihood that a particle was close enough to interact with some given structure in a cell.

## Supporting information

**S1 Text. Detailed description of U-EM.**
(PDF)

**S2 Text. Detailed description of SMC-EM.**
(PDF)

**S3 Text. Gain and covariance simulation based on statistical data.**
(PDF)

**S4 Text. Probability density function of measurements considering pixel dependent read-out noise.**
(PDF)

**S5 Text. Analytical approximation of MLE$_{sCMOS}$.**
(PDF)

**S1 Video. A typical video showing the relationship among trajectory, observation, and properties of readout noise brought by sCMOS.**
(PDF)

**S1 Fig. Computation time record for different SPT algorithms.**
(PDF)

## Author Contributions

**Conceptualization:** Sean B. Andersson.

**Formal analysis:** Ye Lin.

**Funding acquisition:** Sean B. Andersson.

**Investigation:** Ye Lin.

**Methodology:** Sean B. Andersson.

**Project administration:** Sean B. Andersson.

**Software:** Ye Lin.

**Visualization:** Sean B. Andersson.

**Writing – original draft:** Ye Lin.

**Writing – review & editing:** Sean B. Andersson.

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
