## [Decision Letter · Decision Letter 0]

15 Feb 2021

PONE-D-20-35561

Expectation maximization based framework for joint localization and parameter estimation in single particle tracking from segmented images

PLOS ONE

Dear Dr. Andersson,

Thank you for submitting your manuscript to PLOS ONE. After careful consideration, we feel that it has merit but does not fully meet PLOS ONE’s publication criteria as it currently stands. Therefore, we invite you to submit a revised version of the manuscript that addresses the points raised during the review process.

Please find below the reviews for your paper. Please submit a point by point response to the reviewer's comments and highlight the changes in color to facilitate assessing the differences.

We look forward to receiving your revised manuscript.

Kind regards,

Antonio Agudo

Academic Editor

PLOS ONE

Journal Requirements:

2.In your Data Availability statement, you have not specified where the minimal data set underlying the results described in your manuscript can be found. PLOS defines a study's minimal data set as the underlying data used to reach the conclusions drawn in the manuscript and any additional data required to replicate the reported study findings in their entirety. All PLOS journals require that the minimal data set be made fully available. For more information about our data policy, please see http://journals.plos.org/plosone/s/data-availability.

Additional Editor Comments:

Dear authors,

Both reviews consider the paper contain good points for publication. They did a great work. However, both still have several concerns to be addressed. Please address the comments and requests point by point.

Best

Reviewers' comments:

Reviewer's Responses to Questions

**Comments to the Author**

1. Is the manuscript technically sound, and do the data support the conclusions?

Reviewer #1: Yes

Reviewer #2: Yes

2. Has the statistical analysis been performed appropriately and rigorously? 

Reviewer #1: Yes

Reviewer #2: Yes

3. Have the authors made all data underlying the findings in their manuscript fully available?

Reviewer #1: No

Reviewer #2: Yes

4. Is the manuscript presented in an intelligible fashion and written in standard English?

Reviewer #1: Yes

Reviewer #2: Yes

5. Review Comments to the Author

Reviewer #1: This paper proposes a method to jointly estimate the location of a single particle which diffuses on a media with unknown diffusion parameters. Assuming previously segmented images, the method iteratively combines a classical 2D position tracker with an ME estimator for the diffusion coefficients. Since motion depends on these coefficients, it is reasonable to estimate localization and parameters jointly, which is the role of the iterative loop.

Both the particle size and the motions to estimate are in the order of the nm and thus the paper focuses greatly on the effect that the different sources of noise have on final estimate.

The methods are tested in simulation only.

The paper is well written and for the most part it follows easily. However, I have been surprised by the organization of the presented material. This is possibly due to the editor’s constraints, but I do not see why the methods are coming as separate material labelled “supplemental”. I understand that if an article is about studying a particular phenomenon, particularly in the Natural Sciences, then the methods can be assumed supplemental (in case these methods are well-known or at least published elsewhere). But the present paper is about the methods themselves, since all the analyzed data is simulated and therefore do not reflect any real phenomenon (only realistic). In my opinion, S1 and S2 at least should go in the main body, and they should be the basis upon which to base our evaluation.

The key contributions can be divided in two parts. First, there are the methods. Second, there is a detailed benchmark of these methods compared with some predecessors as they have been introduced in the state of the art sections.

The paragraph above about the organization of the paper is pertinent for the first part, that is, the presentation of the methods. Although the methods are correct and seem to improve on the state of the art. Here I assume the state of the art is that of the particular discipline of particle tracking in biological macromollecules, for which I am not competent. If this state of the art is correct, then I must say that the community needs to make an effort because there is a lot of room for improvement (see a few paragraphs below). Concerning this paper, I think the presented methods fall short in some aspects, discussed below.

First, the model parameters to estimate are very simplistic. Having a medium with anisotropic diffusion, the pair Dx, Dy is only pertinent if one aligns perfectly the material with the reference frame. I would have liked to see at least a third parameter estimated, that is the relative orientation of the medium with respect to the reference frame, or equivalently, a covariance matrix representing this diffusion as e.g. D = [Dxx, Dxy ; Dxy, Dyy] (or similar). In case of aligned frames we’d have Dxy=0, Dxx=Dx^2, Dyy=Dy^2.

Second, the UKF has some design parameters that are worth exploring. In particular, one chooses \\alpha in the range [0,1] (S1, paragraph after eq. 3) to decide how the control points are located around the mean to approximate the covariance. In the results section, the authors state that U-EM does not perform well in some cases due probably to a deficient sampling, which makes this paramter \\alfa very pertinent. Since the methods are just presented quickly and fall in the supplemental material, they are poorly discussed. The reader is left with the impression that better results could be obtained, or otherwise that a proper discussion of the limits of the method (UKF in this case) is missing.

Third, the estimation problem posed in this paper seems very simple. In the area of robotics, for example, we are used to estimate the location of a robot in space, together with many other parts of the system (location of objects in the environment, sensor calibration parameters, velocities and acceleration, time delays), all this in 3d (that is, with 6DoF). These works deal with nonlinear systems with hundreds or thousands of variables, while the method presented here only estimates four parameters, and only for a single particle. The estimation literature also contains works to jointly estimate the covariance of the perturbations to the system, which in the present paper are the diffusion parameters. I would recommend the readers to have a look at the robotics literature on simultaneous localization and mappping (a term which reminds the title of the present paper), for example using EKF (Davison 07), UKF (Sotoodeh Bahraini 15), PF (Thrun 04), (Campi 08, this one used not in robotics but in magnetoencephalography), or factor graphs (Kaess 11).

For the second part of the contributions, that is the benchmarking of results, I have to say that I found it very interesting, for it is my first contact with experiments in the nano-scale. Although I cannot judge it, I found the discussion on the error models interesting. I have two remarks. The first one would be that of the tuning of the UKF, as said above, which if explored could have changed the panorama of the results presented, perhaps only slightly but anyway worth exploring. The second remark concerns the observable estimation bias clearly visible in Figs. 5 and 6, where the true values seem to be always at the top of the third quartile, sometimes slightly outside of it. Again, being an article about the estimation methods themselves, a commentary here I feel is absolutely necessary. Often, such biases appear due to nonlinearity, but can also appear due to non-Gaussian noise. Both causes are taken care of by UKF and PF, and therefore one would ideally expect these biases to be absent.

The rest of my comments are minor (typos and the like, small questions and improvements for clarity):

1. Fix this: “Q(θ(e+1), θ(e)) ≥ Q θ(e), θ(e)

ensures the EM algorithm converges to a local *minimum of the likelihood function.” —> local maximum/optimum of the likelihood function

2. In S1, (13,14,15), is T=N? Is this a typo?

3. PSF in (13), maybe center the distribution at (x0,y0) as was done in (1).

4. Fig 5 (right) use equal X and Y scales

5. Fig 7: use better a logscale for G, e.g. 1,2,5,10,20,50,100 or finer, but not a two-piecewise linear scale as shown.

6. Fig 7: label the color scale (color corresponds to…) and a percentage is not [0,1] but [0,100]

7. End of page 10. Add a dot after “G=3” in “at an intensity of G = 3 With an SBR of 10”.

8. Fig 9 I’d prefer (nm) and not (um) in the Y scales, to relate them to previous figures in the text which are in nm.

9. Fig 9 caption: add a dot after “(dashed line)”

10. Results case 3: written Fig *9 —> must be Fig *10 !!

11. Does keeping Dt/dt = constant make sense? At lower shutter time dt, there might be no image quality at all?

12. Many other places missing a dot at end of sentence. e.g. “low (G = 10) and one high (G = 100) A typical image frame” in page 12. Please check thoroughly.

13. Discussion and *Ccnclusion —> Conclusion

Video: not clear what are the axes of the pixellated images. Do they cover the whole area of the figure top-left? If yes, this is weird since in top-right the particle seems to be always in the center. If not, then please specify.

——————————————————————————————

References for this review:

[Davison 07] Davison, Andrew & Reid, Ian & Molton, Nicholas & Stasse, Olivier. (2007). MonoSLAM: real-time single camera SLAM. IEEE transactions on pattern analysis and machine intelligence. 29. 1052-67. 10.1109/TPAMI.2007.1049.

[Sotoodeh Bahraini] Sotoodeh Bahraini, Masoud & Bozorg, Mohammad. (2015). SLAM Using UKF with Adaptation of Scaling Parameter.

[Thrun 04] Thrun, Sebastian & Montemerlo, Michael & Koller, Daphne & Wegbreit, Ben & Nieto, Juan & Nebot, Eduardo. (2004). FastSLAM: An Efficient Solution to the Simultaneous Localization And Mapping Problem with Unknown Data. Journal of Machine Learning Research. 4.

[Campi] Campi, Cristina & Pascarella, Annalisa & Sorrentino, Alberto & Piana, Michele. (2008). A Rao–Blackwellized particle filter for magnetoencephalography. Inverse Problems. 24. 025023. 10.1088/0266-5611/24/2/025023.

[Kaess 11] Kaess, Michael & Ranganathan, Ananth & Dellaert, Frank. (2009). iSAM: Incremental Smoothing and Mapping. Robotics, IEEE Transactions on. 24. 1365 - 1378. 10.1109/TRO.2008.2006706.

Reviewer #2: In this article, the authors highlight simultaneous identification and diffusion coefficient determination for the single particle tracking, with an emphasis on 2-D diffusive systems. Two different methods were introduced: Sequential Monte Carlo-Expectation Maximization (SMC-EM) and Unscented Kalman Filter (UKF) . They envisioned that EM based methods outperform the standard techniques, especially at low signal levels. Their algorithm operates at low and high diffusion rate, at low and high signal level and in two different cameras with different noise sources. They have thoroughly discussed the performances of the algorithms under various conditions. The authors also attempted to extend their algorithm to 3-D systems. This reviewer found this manuscript interesting and would recommend its acceptance after some minor revision.

1. The authors attempted to overcome the coupling problem for simultaneous localization and parameter estimation. To get success, they initially developed Sequential Monte Carlo-Expectation Maximization method for nonlinear system tracking which they published in 2015 (ref. 20). Due to the computational complexity the authors again modified the above method and address the Unscented Kalman filter (UKF) and an Unscented Rauch-Tung-Striebel smoother (URTSS). They also published this work in 2019 (ref. 21). The only new approach here they compare their method with standard method (e.g., GF-MSD and GF-MLE), range of signal to noise ratio, two different camera methods with different noise sources, and low and high diffusion rate. It would be great if the authors could emphasize the novelty of the current work and better distinguish it from the previous ones.

2. They separately defined the method for slow and fast particle diffusion. If the rate of particle diffusion in a sequence of images can differ with time, how can SPT be analyzed or is it accurately possible by these methods?

3. Some minor edits:

- line 7: ‘been’ written double times

- Line 383: Spelling mistake ‘conclusion’

6. PLOS authors have the option to publish the peer review history of their article (what does this mean?). If published, this will include your full peer review and any attached files.

Reviewer #1: No

Reviewer #2: No

---

## [Author Response · Author response to Decision Letter 0]

19 Mar 2021

Please see our detailed Response to Reviewers Letter.

---

## [Decision Letter · Decision Letter 1]

3 May 2021

Expectation maximization based framework for joint localization and parameter estimation in single particle tracking from segmented images

PONE-D-20-35561R1

Dear Dr. Andersson,

We’re pleased to inform you that your manuscript has been judged scientifically suitable for publication and will be formally accepted for publication once it meets all outstanding technical requirements.

Kind regards,

Antonio Agudo

Academic Editor

PLOS ONE

Additional Editor Comments (optional):

All the reviewers' comments were addressed properly. Now, the paper is ready for publication.

Reviewers' comments:

Reviewer's Responses to Questions

**Comments to the Author**

1. If the authors have adequately addressed your comments raised in a previous round of review and you feel that this manuscript is now acceptable for publication, you may indicate that here to bypass the “Comments to the Author” section, enter your conflict of interest statement in the “Confidential to Editor” section, and submit your "Accept" recommendation.

Reviewer #1: All comments have been addressed

Reviewer #2: All comments have been addressed

2. Is the manuscript technically sound, and do the data support the conclusions?

Reviewer #1: Yes

Reviewer #2: Yes

3. Has the statistical analysis been performed appropriately and rigorously? 

Reviewer #1: Yes

Reviewer #2: Yes

4. Have the authors made all data underlying the findings in their manuscript fully available?

Reviewer #1: Yes

Reviewer #2: Yes

5. Is the manuscript presented in an intelligible fashion and written in standard English?

Reviewer #1: Yes

Reviewer #2: Yes

6. Review Comments to the Author

Reviewer #1: Thank you for the clear answers to my comments. You have convinced me on the pertinence of your choices regarding this kind of research. I encourage you to have a look at the literature on estimation existing for localization and mapping in robotics, it is vast and powerful.

Reviewer #2: The authors have adequately addressed all of the previous concerns. I recommend the acceptance of this manuscript by PLOS ONE.

7. PLOS authors have the option to publish the peer review history of their article (what does this mean?). If published, this will include your full peer review and any attached files.

Reviewer #1: **Yes: **Joan Solà

Reviewer #2: No

---

## [Editor Report · Acceptance letter]

12 May 2021

PONE-D-20-35561R1 

Expectation maximization based framework for joint localization and parameter estimation in single particle tracking from segmented images 

Dear Dr. Andersson:

I'm pleased to inform you that your manuscript has been deemed suitable for publication in PLOS ONE. Congratulations! Your manuscript is now with our production department. 

Kind regards, 

on behalf of

Dr. Antonio Agudo 

Academic Editor

PLOS ONE